# Vestibular Assessment with the vHIT and Skull Vibration-Induced Nystagmus Test in Patients with Nonprogressive Vestibular Schwannoma

**DOI:** 10.3390/jcm13092454

**Published:** 2024-04-23

**Authors:** Ioana Brudasca, Gabrielle Vassard-Yu, Maxime Fieux, Romain Tournegros, Olivier Dumas, Georges Dumas, Stéphane Tringali

**Affiliations:** 1Hospices Civils de Lyon, Centre Hospitalier Lyon Sud, Service d’ORL, d’otoneurochirurgie et de Chirurgie Cervico-Faciale, CEDEX, 69310 Pierre Bénite, Francestephane.tringali@chu-lyon.fr (S.T.); 2Université de Lyon, Université Lyon 1, 69003 Lyon, France; 3UMR 5305, Laboratoire de Biologie Tissulaire et d’Ingénierie Thérapeutique, Institut de Biologie et Chimie des Protéines, CNRS/Université Claude Bernard Lyon 1, 7 Passage du Vercors, CEDEX 07, 69367 Lyon, France; 4Department of Oto-Rhino-Laryngology Head and Neck Surgery, University Hospital, 38043 Grenoble, France; georges.dumas10@outlook.fr; 5Research Unit 3450 DevAH-Development, Adaptation and Handicap, Faculty of Medicine, University of Lorraine, 54500 Vandoeuvre-lès-Nancy, France

**Keywords:** video head impulse test, skull vibration-induced nystagmus test, vestibular schwannoma, vestibular disorders

## Abstract

**Background**: Our primary objective was to monitor nonprogressive unilateral vestibular schwannomas (VSs) to assess the efficiency of rapid bedside examinations, such as the video head impulse test (vHIT) and skull vibration-induced nystagmus test (SVINT), in identifying vestibular damage. **Methods**: An observational study was conducted from March 2021 to March 2022 on all adult patients (>18 years old) with a confirmed nonprogressive VS (no active treatment). The SVINT (using a 100 Hz vibrator with two (SVINT2) or three (SVINT3) stimulation locations) and vHIT (for the six semicircular canals (SCCs)) were performed on all patients. The asymmetry of function between the vestibules was considered significant when the gain asymmetry was greater than 0.1. Rapid and repeatable assessment of VSs using two- and three-stimulation SVINT plus vHIT was performed to quantify intervestibular asymmetry. **Results**: SVINT3 and SVINT2 triggered VIN in 40% (24/60) and 65% (39/60) of patients, respectively. There was significant asymmetry in the vestibulo-ocular reflex (VOR), as shown by a VS-side gain < healthy-side gain in 58% (35/60) of the patients. Among the patients with significant gain asymmetry between the two vestibules according to the vHIT (VS-side gain < healthy-side gain), the proportion of patients expressing vestibular symptomatology was significantly greater than that of patients without any symptoms [67% (29/43) vs. 35% (6/17), respectively; *p* = 0.047]. **Conclusions**: The SVINT2 can be combined with the vHIT to form an interesting screening tool for revealing vestibular asymmetry. This work revealed the superiority of mastoid stimulation over vertex stimulation for SVINT in patients with unilateral vestibular loss.

## 1. Introduction

The skull vibration-induced nystagmus test (SVINT) is a vestibular test that allows visualization of compensated or noncompensated vestibular and/or otolith asymmetry by provoking horizontal nystagmus during vibratory stimulation [1]. This test is rapid, noninvasive, repeatable and allows visualization of all vestibular deficits, even the compensated ones [2,3,4,5]. In previous studies on patients with VS, vibration-induced nystagmus (VIN) was present in between 44% and 78% of patients [3]. The video head impulse test (vHIT) is another noninvasive and repeatable test that has been widely used and validated [3,5] for measuring and mapping deficits in the semicircular canals (SCCs), even when asymmetry is compensated [6,7]. These two tests are complementary since the vHIT analyzes only the response of the six SCCs, and the SVINT is a more global test that corresponds mainly to the response of the horizontal SCC but also of the superior SCC and of the utricle [8,9,10,11]. Compared with those of healthy individuals, all three SCC functions can be significantly altered on the monitored side of the VS, even on nonprogressive sides [12].

Our primary objective was to describe the results of the rapid assessment of vestibular damage with the vHIT and SVINT in patients with monitored, nonprogressive unilateral VS. Our secondary objective was to investigate the correlation between these findings, MRI features, and audiometric data. The third objective was to compare mastoid stimulation over vertex stimulation in the SVINT.

## 2. Materials and Methods

### 2.1. Design and Ethics

This observational study was conducted from March 2021 to March 2022 at a tertiary hospital specializing in oto-neurosurgery. The inclusion criteria were adult age (>18 years old), confirmed, nonprogressive VS based on axial or volumetric measurement of the lesion for at least twelve months (based on the comparison of the last two MRI scans (12 or 24 months apart) during a multidisciplinary skull base meeting), absence of active treatment (surgery or radiotherapy), and no medical history of vestibular pathology (hydrops, history of neuritis, benign paroxysmal positional vertigo, vestibular migraine, labyrinthine fistula) or neurological pathology responsible for balance disorder prior to VS diagnosis. Patients with bilateral VS or neurofibromatosis were excluded.

This study was conducted in accordance with the Declaration of Helsinki and was approved by the institutional review board (approval n° 22-5048). Informed consent was obtained from all participants. The data collected were strictly anonymized.

### 2.2. Population

The data collected included general demographics (age, sex), VS side, and date of diagnosis to determine the follow-up duration. VSs were classified according to the Koos classification on the latest available MR imaging (1.5T or 3T, T2 high-resolution (HR)/constructive interference in the steady-state (CISS)/fast imaging employing steady-state acquisition (FIESTA) sequences) [13]. Each patient underwent an audiometric test in a soundproof booth, and the results for the pathological ear were classified according to the American Academy of Otolaryngology—Head and Neck Surgery (AAO-HNS) scale [14] into useful (classes A and B) and nonuseful (classes C and D) hearing. The patient’s current cochlear andvestibular symptomatology (vertigo, instability, hypoacusis, tinnitus) was collected during an interview in a declarative yes/no manner. Then, all patients underwent the SVINT and vHIT, which were performed by ENT doctors (3 examiners in total) and a vestibular physiotherapist.

### 2.3. Vestibular Functional Investigations: The SVINT

The SVINT was performed by using a vibrator at 100 Hz stimulation (Vibrateur Vestibulaire—VVIB 3F, Synapsys, Marseille, France). This stimulator is a mechanical vibrator with a vibration amplitude of 1 mm and a cylindrical contact surface 2 cm in diameter. The circular contact surface was covered with thin rubber. The examiner stood in front of the seated patient and applied the vibrator perpendicular to the bone surface for 5 to 10 s per site from one hand. The examiner used the other hand to maintain and immobilize the patient’s head. The nystagmus was recorded under a videonystagmoscope (Nystagview, Synapsys, Marseille, France) while vibration was applied and we focused on the horizontal nystagmus. The vibrator was applied both on the (left or right) mastoid, behind the ears at the external acoustic meatus level, and on the vertex right above the ears in the sagittal plane. For the mastoid vibration, the examiner avoided the mastoid tip to prevent muscular vibration radiation and proprioceptive involvement. If the VIN appears instantly when the vibrations start and disappears when the vibrations stop, then only a few seconds of vibration per bone location are enough to collect the information.

Two results were collected: the VIN with stimulation at 3 locations (SVINT3, applied on both mastoids and the vertex) and the VIN with 2 locations (SVINT2, applied only on the two mastoids). For each application, the SVINT was considered positive when it triggered nystagmus beating with a maximal slow-phase velocity (SPV) greater than 2°/s in the same direction for each of the stimulations [5]. The examiner performed a qualitative analysis of the nystagmus, and the data were summarized as the presence or absence of same direction VIN regardless of the site of vibratory stimulation. The SVINT was performed by using a vibrator at 100 Hz stimulation (Vibrateur Vestibulaire—VVIB 3F, Synapsys, Marseille, France) applied perpendicular to the bone surface for 5–10 s per site under a videonystagmoscope (Nystagview, Synapsys, Marseille, France).

### 2.4. Vestibular Functional Investigations: The vHIT

The vHIT for the six SCCs was recorded with a vHIT device (vHIT Ulmer, Synapsys, Marseille, France) composed of an infrared video camera on an adjustable tripod and a computer with vHIT software (CEI 62304, CEI 82304-1, Synapsys, Marseille, France).

The patient was seated in front of the infrared camera at a distance of 90 cm from the camera and was asked to maintain their gaze on a visual target placed beyond and above the camera at eye level. The target was at a distance of two meters from the eyes of the patient. The infrared camera recorded head and eye positions in response to passive head impulses with a small range of motion (inferior to 20° from the neutral position) and a high velocity applied to the head of the patient by the examiner. The examiner stood behind the patient and manually moved the patient’s head to the planes of the six SCCs at high speed and within 20° of the neutral position. The order of the direction of head impulses was randomized. The velocity threshold of the head impulses was 200°/s or more for the anterior and posterior SCCs and 250°/s or more for the lateral SCC. For each SCC, at least five head impulses were recorded. The vHIT software provided auditory and visual feedback to help the examiner assess the velocity threshold and maintain the head of the patient in the right plane.

The gain of the vestibulo-ocular reflex (VOR) for each canal was calculated by the algorithm of the vHIT software for each head impulse. The final value of VOR gain used for the statistical analysis was the mean calculated by the vHIT software for each SCC after at least five valid acquisitions, with a standard deviation strictly less than 0.10. The examiner had the ability to manually erase outlier measures or those that did not seem accurate. The VOR gains were considered normal if they were greater than 0.8 for the lateral SCCs and 0.7 for the anterior and posterior SCCs [15]. In addition to the VOR gain, the vHIT software was used to calculate the level of asymmetry between the left and right vestibules. An asymmetry of function between the vestibules was considered significant if the gain asymmetry was greater than 0.1 [8,12].

### 2.5. Statistical Analysis

The main outcome measure was the rapid and repeatable vestibular assessment of nonprogressive monitored VS (nonactive treatment) using the SVINT2 and SVINT3 plus vHIT to quantify intervestibular asymmetric responses. Categorical variables are summarized as proportions (numbers), and continuous variables are summarized as medians and interquartile ranges. The distribution of the variables was, for the most part, asymmetrical; the data of a small number of participants did not allow any approximation by a normal distribution. Comparisons between groups (with or without vestibular symptoms) were made using Fisher’s exact test for categorical variables. Correlations (between vestibular test results and vestibular symptoms, MRI Koos staging, and audition impairment) were identified with the chi^2^ Pearson test. The threshold for statistical significance was set at 0.05. Analyses were performed with the R software (v. 4.1.2, R Foundation for Statistical Computing, Vienna, Austria, www.r-project.org accessed on 6 March 2024).

## 3. Results

A total of 60 patients were included, 34 women and 26 men, with a median age of 68 years and a median follow-up of 72 months. Sixty-one percent (37/60) of VSs were on the left side, and examples of VSs are shown in Figure 1. Among the patients, 72% (43/60) described vestibular symptoms such as instability [56% (34/60)] and/or rotatory vertigo [53% (32/60)]. All details are in Table 1.

The SVINT3 triggered the same direction of VIN in all three stimulations in 40% of patients, with a beat toward the healthy ear in 20%. A total of 41% of the symptomatic patients had VIN, and 37% were in the asymptomatic group; the difference between the two groups was not statistically significant [41% vs. 37%, respectively; *p* = 0.5].

The SVINT2 triggered a VIN in 65% of patients, with 41% demonstrating beating toward the healthy ear. A total of 68% of the symptomatic patients had VIN, and 50% were in the asymptomatic group. There was no statistically significant difference in SVINT3 expression between the two groups [68% vs. 50%, respectively; *p* = 0.19].

The vHIT showed one or more decreased VOR gains in 76% (46/60) of the patients on the VS side. The VOR gains were decreased for the posterior SCC in 71% (42/60), for the lateral SCC in 41% (25/60), and for the anterior SCC in 20% (12/60) of patients. On the healthy side, 52% (32/60) had at least one abnormal VOR gain, 45% (27/60) in the posterior SCC, 30% (18/60) in the lateral SCC, and 10% (6/60) in the anterior SCC. There was significant asymmetry in the VOR gains, with VS-side gain < healthy-side gain in 58% (35/60) of the patients. Among the patients with significant gain asymmetry between the two vestibules according to the vHIT (VS side < healthy side), the proportion of patients expressing vestibular symptomatology was significantly greater than the proportion without any symptoms [67% (29/43) vs. 35% (6/17), respectively; *p* = 0.047]. The distribution of VOR gains for each SCC is plotted for the VS side (Figure 2A–C) and the healthy side (Figure 2D–F). Combining the two tests, 48% (29/60) of the patients in our study had significant SVINT2 VIN and a deficit in at least one SCC on the vHIT; while 90% (54/60) had one and/or the other.

There was no statistically significant difference between the groups regarding those with a test revealing a vestibular deficit (pathological vHIT on the VS side and positive SVINT2) and those with normal results on one or both vestibular tests for vestibular symptoms, Koos stage, or hearing impairment. Sixty-two percent (18/29) of the patients who showed a vestibular deficit in the vestibular tests had Koos stage 1 VS, and thirty-eight percent (11/29) had Koos stage 2 VS. Among the patients with normal vHIT or SVINT2, fifty-five percent (17/31) had Koos stage 1, and forty-five percent (14/31) had Koos stage 2. There was no significant correlation between the test results and MRI Koos staging (*p* = 0.76) (Table 2). Sixty-two percent (18/29) of the patients with a vestibular deficit in the tests had class A–B hearing on the VS side, and thirty-eight percent (11/29) had class C–D hearing. Among the patients without vHIT- or SVINT2-detected vestibular deficits, 60% (21/31) had class A–B hearing on the VS side, and 32% (10/31) had class C–D hearing. There was no significant correlation between the test results and the AAO-HNS hearing class (*p* = 0.85) (Table 2).

## 4. Discussion

### 4.1. Main Results

The SVINT is a very practical tool to use, and the presence of vibration-induced nystagmus is a useful indicator of intervestibular asymmetry. The SVINT is more useful for peripheral disorders than for central disorders [3]. The usefulness of SVINT for pathologies other than VS, such as Meniere’s disease, vestibular neuritis, and Minor’s syndrome, has already been demonstrated [3]. The test has good reproducibility, and there is a small habituation effect when the test is repeated [4]. Even if the patient no longer has vestibular symptoms, the SVINT may reveal “vestibular scarring” of past vestibular events that might have been compensated for (“scarring” here refers to irreversible damage to the vestibular function by the underlying pathology). Moreover, in the context of VS, the SVINT is a good indicator of vestibular asymmetry before and after surgery [2,3]. In our study, we found that the SVINT2 revealed vestibular asymmetry in a greater number of patients than the SVINT3 did, as reported in a recent review of the literature [16]. This finding reinforces the rapid nature of this test, in addition to its relevance in the follow-up of VS patients.

The main aim of this study was to describe the results obtained for patients with nongrowing VS using two tests that can be performed rapidly at the bedside or in an ENT follow-up consult: the vHIT and the SVINT. To our knowledge, this is the first study to investigate the combined use of the SVINT and vHIT to monitor small, nontreated VS. We observed that 40% of the patients with VS had VIN using the SVINT3, while 65% had VIN triggered with the SVINT2. This finding is consistent with other studies in the literature showing that vertex vibration might be a less effective stimulation than mastoid vibration [3,4]. Our SVINT2 results are similar to those of other series showing a sensitivity of 45–65% in nonoperated VS [2]. Lee et al. showed a significant correlation between intervestibular asymmetry and the speed of vibration-induced nystagmus (VIN) by comparing slow-phase VIN before and after trans-labyrinthic surgery [1]. Clinically, we were unable to find a correlation between patients’ vestibular symptoms and SVINT test results, but as stated earlier, this was expected, as the SVINT permits the revealing of past vestibular compensation events.

### 4.2. Decreased VOR Gain on the VS Side

Using the vHIT, we detected a significantly lower mean VOR gain on the homolateral side to the VS for 74% of the patients. These results echo those previously reported [7,17]. West et al. reported an 80% sensitivity of the vHIT in unilateral VS [18]. However, previous studies did not include an additional control group with similar symptoms but negative MRI findings. Taylor et al. included patients with VS and symmetrical hearing (21.4% of the study population), which may explain some of the differences from our study regarding the prevalence of vestibular dysfunction [19]. In addition, Fujiwara et al. reported that the vHIT was a useful tool for diagnosing dysfunction of the semicircular canals, including the vertical canals [17]. As similar results have been obtained in several reported series [7,20], we can conclude that the vHIT is a good diagnostic test for assessing vestibular function in patients with VS. Most of our patients had posterior SCC VOR loss (71%), followed by damage to the lateral SCC (41%), and finally, to the anterior SCC (20%), which is consistent with the findings of Khrais et al., who showed that 91% of VSs originate from the inferior vestibular nerve [21]. However, a weak but significant association between deterioration in vHIT gain and the patient’s vertiginous complaint was found (*p* = 0.0047), which was not the case in the Tranter-Entwistle et al. smaller 30-patient cohort [22]. This weak correlation may be explained by possible selection bias or vestibular compensation over time, as some of our patients were followed for a long time without any recent vestibular symptoms. To avoid asymmetric vestibular function being explained by another vestibular or neurological cause, we did not include patients with symptoms of vertigo prior to VS diagnosis, those with ENT or neurological comorbidities that could affect balance, or those with bilateral vestibular areflexia. In addition to VOR gain, which has been validated as an objective outcome measure, another indicator of a semicircular canal deficit is the presence of catch-up saccades [12]. We did not take this indicator into account in this study, which could also explain the weak association between vestibular complaints and abnormal vHIT results.

### 4.3. Combined vHIT and SVINT

In 2021, Martin-Sanz et al. studied the diagnostic value of the SVINT in association with the vHIT for detecting VS in people with unilateral sensorineural hearing loss (USHL) [8]. A total of 53 people were included in their prospective study. The control group comprised 23 patients with USHL and normal MRI results, and the study group comprised 33 patients with USHL and VS [8]. Researchers have suggested that using the vHIT may be an efficient approach for screening for VS in patients with asymmetric sensorineural hearing loss [8]. These two parameters increased from 78% (0.95–0.61) and 75% (0.90–0.61) for vHIT alone to 95% (1.03–0.87) and 66% (0.82–0.50), respectively, for vHIT in combination with SVINT [8]. The SVINT assessment protocol described in this study differed from the one used in our study. Mastoid vibration was applied on the mastoid process and in the lower part of the sterno-cleido-mastoidian muscle, involving the cervical proprioception response. Therefore, this study showed that the SVINT is an interesting tool for detecting VS in patients with USHL. If the SVINT is a promising tool for the detection of VS, the use of the tool for the follow-up of VS patients also seems interesting. In our study, we found no significant difference between patients with pathological vHIT and positive SVINT2 results and those with normal results for one or both tests regarding vestibular complaints (*p* = 0.22), MRI Koos stage (*p* = 0.39), or AAO-HNS hearing loss stage (*p* = 0.59). This may be explained by another selection bias inherent in our study. The included patients had small VS, only grade 1 and 2 according to the Koos classification (with most Koos stage 1 (58%)), whereas Martin-Sanz et al. found a significant correlation between the Koos stage and test (SVINT and vHIT) results in their 33 Koos stage VS population [8]. In their cohort of 69 patients, Blödow et al. showed a correlation between caloric deficit and tumor volume, which was not found with the vHIT [23]. Batuecas-Caletrio et al. showed a weak correlation (with a correlation coefficient of 0.54) between canal deficits on the vHIT and tumor volume for Koos stage 3 and 4 tumors only, with 90% of VOR gains lowering on the horizontal canal, compared with 41% in our population [24]. Therefore, SVINT and vHIT cannot replace active MRI monitoring of VS growth. We also know that patients’ functional complaints depend not only on vestibular test results but also on anxiety and depression symptoms, disease perception, and coping strategies [6]. Concerning hearing loss, we did not find any correlation between the AAO-HNS hearing loss stage and the vHIT and/or SVINT results. West et al. reported a weak correlation (R = 0.3) between vHIT results and VS audiometric findings [18]. However, this might be related to our population of only small-Koos-stage VSs. Currently, vestibulo-audio-volumetric correlations are debated in the literature, with correlations either absent [5] or weak [12,18,23], leaving the field open to research.

### 4.4. Decreased VOR Gain on the Healthy Side

Interestingly, quite a number of our patients presented decreased VOR gains at the vHIT on the healthy side (52%), with a deficit mostly on the posterior SCC (45%) and/or a VIN beating toward the side of the VS (20% in SVINT3 and 24% in SVINT2). This finding has already been reported by Batuecas-Caletrio et al. [24], who reported an 8% deficit on the healthy side in the vHIT. This difference in proportion between the two series could be explained by vHIT testing of the lateral SCC only in the Batuecas-Calterio et al. series and by the younger age of their patients: 52 vs. 68 years. The deficit on the non-VS side might be explained by presbyvestibulia, which is clinically consistent with the median age of 68 years and the preferential involvement of the posterior SCC in our series. A history of other vestibular deficits that may have been wrongly attributed to VS and not investigated cannot be completely ruled out. Finally, internuclear neural connections capable of central inhibition on the healthy side exist to minimize intervestibular asymmetry and should not be overlooked [25].

### 4.5. Strengths and Limitations

This study shows the results of quite a large cohort for the topic of VS, which is still a rare entity even though its diagnosis has increased with easier access and better resolution of MR images. We believe that it is necessary to use SVINT2 in combination with vHIT in an even larger sample of patients to confirm our observations and, if possible, in further studies to include all Koos stages. Regarding, the limitations of our study, they are inherent to its design, with patients assessed at a point in time that does not reflect a potential vestibular evolution during monitoring. Nilsen et al. showed that vestibular symptomatology and vestibular deficits on caloric tests remained stable over time in the absence of progression on MRI [26], which would justify repeated exploration only in cases of MRI or clinical changes. The eventuality of presbyvestibulia or unknown comorbidities that could impact balance and vertigo symptoms should also be discussed, as the median age is 68 years old. The use of standardized questionnaires to assess symptomatology could have enabled a more detailed analysis of the correlation between symptom intensity and canal hyporeflexia and/or VIN intensity. Nevertheless, a yes/no declarative interview saved time and enabled us to integrate this vestibular assessment protocol into a standard follow-up ENT consultation. We estimated the time taken to perform these two tests, including the setup and interpretation, to be 10 min on average. They rapidly provided objective and localizing information on vestibular deficits secondary to VS without ignoring vestibular events that may occur on the “healthy” side. The test results can help clinicians with vestibular follow-up and/or for vestibular physical therapy.

## 5. Conclusions

In our population of 60 patients with nonprogressive unilateral monitored VS, the SVINT2 revealed more vestibular asymmetry than did the SVINT3 (65% vs. 40%), confirming the greater sensitivity of mastoid stimulation than of vertex stimulation. Therefore, we recommend the use of the SVINT2 to screen for intervestibular asymmetric responses in the VS population. Patients with vestibular symptoms were significantly more likely to have vHIT-detected asymmetry [67% (29/43) in symptomatic patients vs. 35% (6/17) in vestibular asymptomatic patients; *p* = 0.047], which confirms that the vHIT is an interesting tool allowing the ENT therapist and the physiotherapist to monitor and quantify the patient’s vestibular symptoms in the field of VS. The combination of the SVINT and vHIT cannot replace MRI for monitoring small VS, as there was no correlation between the vestibular test results and the VS Koos stage. This work newly demonstrated the superiority of mastoid stimulation over vertex stimulation in the SVINT.

## Figures and Tables

**Figure 1 jcm-13-02454-f001:**
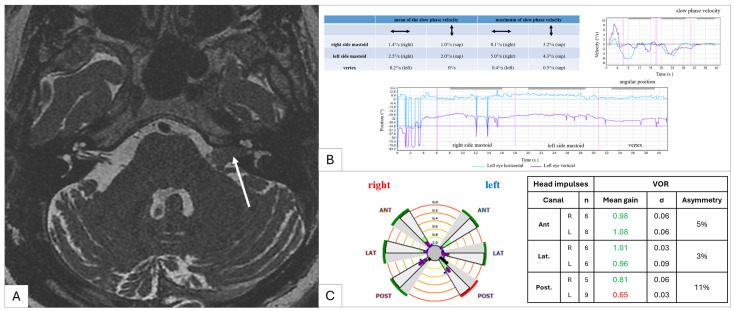
Clinical example of a left vestibular schwannoma (Koos stage 2, white arrow). (**A**) T2 MR image showing a left vestibular schwannoma (white arrow); (**B**) a vibration-induced nystagmus test result (SVINT 3); (**C**) a vHIT test result.

**Figure 2 jcm-13-02454-f002:**
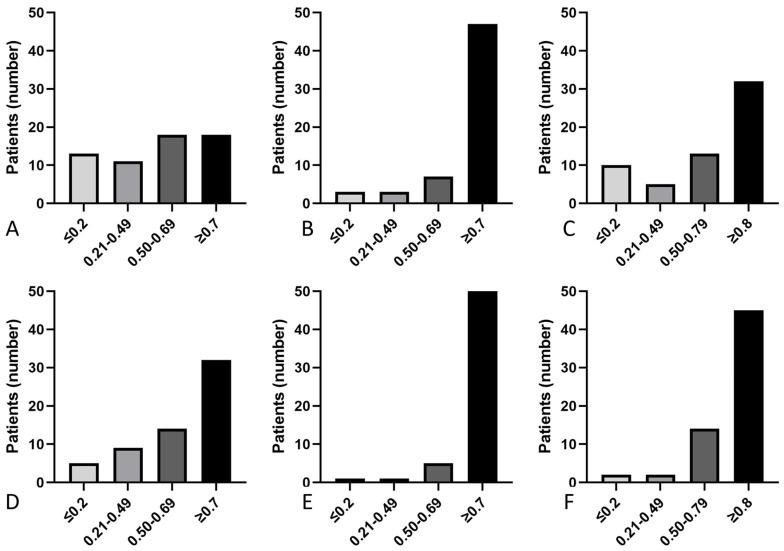
Distribution of VOR gains on the VS side. Distribution of VOR gains on the VS side for the posterior SCC (**A**), anterior SCC (**B**), and lateral SCC (**C**), and on the healthy side for the posterior SCC (**D**), anterior SCC (**E**), and lateral SCC (**F**) with personalized cutoffs. Continuous variables were divided into four groups, ≤0.2, 0.21–0.49, 0.50–0.69, and ≥0.7 for posterior and anterior SCC, and ≤0.2, 0.21–0.59, 0.60–0.79, and ≥0.8 for lateral SCC, both on the VS side and on the healthy side.

**Table 1 jcm-13-02454-t001:** Demographic characteristics of the included population.

Characteristics	n = 60
Sex Male Female	26 (43%) 34 (57%)
Age (years)	68 [60–74]
Duration of follow-up (months)	72 [39–85]
Vestibular symptomatology	43 (72%)
Hypoacusis	49 (81%)
Tinnitus	36 (60%)
Koos classification Stage 1 Stage 2	35 (58%) 25 (42%)
Vestibular schwannoma side Left Right	37 (61%) 23 (39%)
Hearing class (AAO-HNS *) Class A–B Class C–D	39 (65%) 21 (35%)

Values are presented as numbers (proportions) for categorical variables and medians (quartiles) for continuous time variables. * AAO-HNS: American Association of Otorhinolaryngology and Head and Neck Surgery.

**Table 2 jcm-13-02454-t002:** A correlation study between with and without vHIT- or SVINT2-detected vestibular deficits and vestibular symptoms, hearing loss and Koos stage.

	Vestibular Deficit on vHIT and SVINT2 n = 29	vHIT and/or SVINT2 without Vestibular Deficit n = 31	*p* *
Vestibular symptoms			0.119
Yes	24/29 (83%)	19/31 (61%)	
No	5/29 (17%)	12/31 (39%)	
Koos classification			0.76
Stage 1	18/29 (62%)	17/31 (55%)	
Stage 2	11/29 (38%)	14/31 (45%)	
Hearing class (AAO-HNS)			0.85
Class A–B	18/29 (62%)	21/31 (68%)	
Class C–D	11/29 (38%)	10/31 (32%)	

Values are presented as numbers (proportions) for categorical variables. * *p* was considered significant if it was <0.05.

## Data Availability

Data are available on reasonable request to the corresponding author.

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
