# Peer review of "Vestibular Assessment with the vHIT and Skull Vibration-Induced Nystagmus Test in Patients with Nonprogressive Vestibular Schwannoma"

_jcm, 2024, doi:10.3390/jcm13092454_

Round 1

Reviewer 1 Report

Comments and Suggestions for Authors

Major issues

The methods to quantify eye movements during vibration and head impulse is not clearly described and should be clearly articulated—or at least citing previous work where this method is described and normal ranges established.

A figure should show at least one example subject showing their HIT and VIN eye movement traces, and reporting schwannoma size and side.

Minor issues

Line 88 How was the nystagmus quantified and which axes were used? Horizontal, vertical, torsion?

Line 93 Vhit normal gain ranges referenced are for a different instrument. Please describe normal ranges for instrument used.

Line 172 What is vestibular scarring?

Line 222 If mastoid stimulation is recommended rather than vertex, then perhaps the Weber analogy (Line 44) is not so good?

Reviewer 2 Report

Comments and Suggestions for Authors

This article is well-written and clear! I like how the clinical implications are described and how the paper addresses the extant literature. A few suggestions:

Line 107: should be chi2 rather than chi2.

Line 109: Just says "The Materials", not a complete sentence.

- In the first paragraph of the results, you do not need to include the information in Table 1 in parentheticals in the text, because Table 1 is immediately below that paragraph

-Similarly, the paragraph starting at Line 131 could have fewer parentheticals as a result of the well-made figures below.

- The discussion could be broken up to include strengths, limitations, etc.

Reviewer 3 Report

Comments and Suggestions for Authors

I would like to thank the authors for interesting topic and well-structured manuscript. Just some comments:

Introduction seems to be quite short. It can be extended to describe the clinical difference of vHIT and SVINT, why it can be more useful to use both of them as it definitely takes more time to perform two tests instead of only one.

Material and methods are described well but can be improved to provide some details on year 1995 AAJ-HNS scale (please correct the reference, is seems that wrong doi was inserted) as its’ criteria differ from contemporary classification of hearing loss (concerning frequencies and thresholds). This classification is not familiar to me and I’m grateful for the new knowledge in this field.

Results:

Line 114, table 1 - the number of patients with vestibular symptoms is 43 and further in the lines 125, 128, 129, 140 a number of 44 symptomatic patients appears. Are they some other patients or a mistake?

Discussion:

the second paragraph is not clear: line 180 “two tests …: vHIT and VOR” - VOR is not a test, isn’t it? as described earlier in the Methods, line 91-92 “vHIT was performed… to evaluate VOR”

line 211 – mistake? SV instead of VS

What about neurological status of participants? Due to median age of the research cohort 68 (60-78) years, it can be assumed that vestibular symptomatology (especially instability) could be of neurological origin.
